# Mechanical Strength of Waste Materials: A Cone Penetration Testing-Based Geotechnical Assessment for the Reclamation of Landfills

**DOI:** 10.3390/ma18092130

**Published:** 2025-05-06

**Authors:** Marek Bajda, Mariusz Lech, Katarzyna Markowska-Lech, Piotr Osiński, Eugeniusz Koda

**Affiliations:** 1Institute of Civil Engineering, Warsaw University of Life Sciences, 02-787 Warsaw, Poland; mariusz_lech@sggw.edu.pl (M.L.); katarzyna_markowska-lech@sggw.edu.pl (K.M.-L.); p.osinski@unsw.edu.au (P.O.); eugeniusz_koda@sggw.edu.pl (E.K.); 2Faculty of Engineering, University of New South Wales, Sydney, NSW 2033, Australia

**Keywords:** municipal solid waste, mechanical parameters identification, material characteristics, site investigation, experimental testing

## Abstract

The stability and mechanical properties of municipal solid waste (MSW) deposits in closed landfills are critical for safe land reclamation and infrastructure development. This study employs Cone Penetration Testing (CPT) to evaluate the geotechnical parameters of aged waste at three closed landfill sites in central Poland. Key parameters, including shear strength, internal friction angle, density, and liquidity index, were assessed to determine slope stability and bearing capacity for future redevelopment. Due to the heterogeneous nature of MSW, CPT results were analyzed in conjunction with empirical correlations and nomograms to improve accuracy, so the parameters can be used for future numerical modeling and proposing new computational approaches for landfill body elastic and mechanical behavior predictions. The findings indicate significant variability in landfill waste mechanical properties, influenced by waste composition, decomposition stage, and compaction history. The study highlights CPT’s reliable detremination of geotechnical parameters for landfill restoration projects, particularly for infrastructure, creating the potential for green energy and sustainable development. The results contribute to improving engineering practices in landfill restoration and ensuring the long-term stability of post-closure land use. This study also contributes to obtaining reliable results on anthropogenic waste material mechanical parameters at both the material point and at the overall structural scale, benefiting future computational methods and modeling approaches for analyzing structural and geotechnical safety of such complex and demanding structures as landfills.

## 1. Introduction

### 1.1. Geotechnical Challenges and Reclamation of MSW Landfills

Despite progress in the circular economy, landfill sites remain the primary destination for municipal solid waste (MSW). ISWA reports [1] and Bano, 2025 [2] estimate that around 37% of global waste is deposited in controlled, sanitary, or unspecified landfills, while 33% ends up in open dumps. Given the close proximity of landfills to urban areas, concerns persist regarding the environmental and public health risks posed by slope failures, safe reclamation, or the potential for further development of contaminated sites. Ensuring the stability of these sites is a key challenge in environmental geotechnics, as it depends not only on the initial composition and characteristics of MSW but also on long-term degradation processes and operational practices such as compaction, daily covering, leachate and gas drainage, and, in the case of bioreactor landfills, leachate recirculation. When analyzing the stability of such demanding structures or preparing for reclamation planning, the key aspects, such as accurate determination of geotechnical parameters, are crucial for ensuring a safe and stable design. According to the general infrastructure design framework and regulations, landfills are considered civil engineering structures and thus according to European Standards Eurocode 7 [3,4], the process of site investigation when verifying geotechnical safety is divided into three key stages: preliminary research, research for design purposes, and control and monitoring. Comprehensive geotechnical site investigations should encompass field tests, laboratory analyses, additional complementary assessments, and, if required, further inspections and long-term monitoring [4]. The planning of a geotechnical research program should be based on an initial evaluation of the complexity of geological and geotechnical conditions, as well as the type and scale of the intended engineering structure [3].

Considering landfill design and post-closure reclamation projects, they must align with official decisions on land use conditions, the guidelines outlined in local spatial development plans, or the objectives defined in urban and ecophysiographic studies. One of the commonly implemented strategies for landfill site rehabilitation involves repurposing the land for new infrastructure developments, particularly lightweight structures such as photovoltaic farms. This approach not only maximizes the utility of reclaimed land but also supports sustainable energy initiatives and environmental revitalization efforts.

The reliable assessment of mechanical parameters (strength and deformation) of municipal solid waste is essential for slope stability analysis and is the foundation for designing restored landfills. The precise design of structures to be developed on such sites requires computations based on reliable strength parameters (internal friction angle φ, cohesion c, and shear strength) and deformation parameters (e.g., compressibility modulus M, serviceability limit state). Accurate estimation of these parameters enables the selection of an appropriate foundation design approach and ensures the reliability and safety of the designed structures and the landfill body itself.

Determining waste geotechnical parameters is extremely challenging due to the nature of the analyzed material. The composition, heterogeneity, and the degree of decomposition are only a few factors determining waste mechanical performance. The testing of waste should be based on in situ investigations and sampling. Due to the complexity of the factors, as well as the challenges associated with obtaining representative samples of the analyzed material from landfills, laboratory tests may be unreliable or insufficient. Therefore, despite the lack of empirical relationships allowing the estimation of mechanical parameters of deposited waste based on obtained results, the cone penetration test (CPT) is the most effective and reliable method. Due to its nature and the construction of the sensing tip, it allows full understanding of the analyzed waste structure and provides significant possibilities for interpreting selected mechanical parameters of deposited anthropogenic material [5].

### 1.2. Heterogeneity of Waste and Safety of Landfill Body

Before restoring closed contaminated landfill sites, expert evaluations, especially comprehensive geotechnical investigations, are crucial to confirm the engineering feasibility and ensure safety. These assessments help determine ground reinforcement and soil improvement measures to support future construction [6]. Landfill waste is highly heterogeneous, with variations in morphology, density, and mechanical properties [7], complicating accurate geotechnical assessments [8,9]. Due to this complexity, strength parameters must be determined through site-specific in situ investigations and, when possible, verified with laboratory tests of sampled waste [10]. This combined approach enhances reliability in assessing material behavior and supports decision making procedures for reclamation plans. Since waste properties vary by composition, degradation state, and environmental context, designing reclamation works using only literature review waste materials data should be avoided. Regional differences in waste types and quality make direct extrapolation unreliable. Instead, accurate geotechnical data must come directly from the site investigation to ensure structural safety and stability during and after reclamation. Reliable parameters are essential to prevent slope failures, deformations, and safety issues, both during exploitation and post-closure. Knowing mechanical and elastic waste properties enables the use of appropriate constitutive models and advanced computational methods to mitigate civil and environmental risks. While standardized tests can be applied, comparisons should be limited to select properties and waste types [10]. Site-specific parameters ensure more accurate foundation design, accounting for the unique nature of the landfill material. In practice, direct testing of both aged and fresh waste is increasingly common, producing more relevant data for analysis. However, MSW’s variability from natural soils requires tailored testing methods [11]. Among these, in situ testing proves most effective for assessing strength characteristics [12] and should be supported by large-scale laboratory tests [13] and back analysis when applicable. Numerous studies have explored lab testing of waste geotechnical parameters [14,15,16,17,18,19], offering insight into the hydro-thermo-bio-mechanical behavior of MSW [20]. Still, due to the material’s heterogeneity, lab results may not always reflect field conditions.

Waste composition continues to evolve due to changes in consumption (e.g., increased single-use packaging during COVID-19 [21,22,23,24]) and waste treatment efforts [25,26,27,28]. Throughout and after landfilling, MSW change its particle size, density, porosity, and moisture content. These shifts affect mechanical properties like shear strength and stiffness, which vary spatially and over time. Therefore, interpreting test results must consider key factors such as morphology, age, and deposition methods [5,7,12,29,30]. This thorough approach supports safer, more effective engineering for landfill reclamation.

### 1.3. Evaluating the Mechanical Behavior of Aged Waste Using Cone Penetration Tests

Among the most commonly used site investigation methods for geotechnical assessments are the Standard Penetration Test (SPT) and the cone penetration test (CPT) [4,31,32]. However, when applied to municipal solid waste, these methods present unique challenges. In the case of SPT, there is significant uncertainty in deriving strength parameters, as no well-established correlations exist between the blow count and the mechanical properties of waste materials. This limitation makes it difficult to accurately assess the geotechnical characteristics of landfill deposits using SPT alone. This is the major reason why CPT investigation results are much more often recommended to rely on when investigating geotechnical parameters of waste in situ conditions. For CPT, two types of cone penetrometer tips are commonly used in the field: mechanical and electrical. While CPT results are typically interpreted using datasets derived from various soil types, applying these conventional interpretation methods to non-standard materials such as municipal waste is considerably more complex. Figure 1 presents a scheme for conducting the test with the mechanical cone and tip features. Despite these challenges, CPT remains one of the most valuable techniques for assessing restored landfill sites in terms of its mechanical behavior. Static CPT provides the most representative data for analyzing the internal structure and mechanical properties of waste materials.

Proper interpretation of measured parameters such as cone resistance (q_c_), sleeve friction (f_s_), and the friction ratio (R_f_) across different depth profiles allows for the application of existing nomograms (Figure 2). These nomograms facilitate the classification of landfill materials and enable empirical estimations of their strength parameters. As waste degrades, its structural integrity is altered, impacting its geotechnical behavior and overall stability of the landfill. Research findings published in the available literature [6] indicate that the internal friction angle (φ) tends to decrease as waste decomposition progresses. Specifically, due to biodegradation over extended periods (typically 20–30 years of storage), φ can decrease from approximately 40° to 20°. This reduction is likely associated with an increasing proportion of fine particles within the waste matrix, which occurs as larger waste fragments break down during the aging process. Studies conducted by various researchers [8,9,10,31] have demonstrated that municipal waste exhibits a remarkably wide range of strength parameters. However, significant variability in these values can be attributed to differences in testing methodologies, as there is currently no standardized approach for geotechnical waste analysis. The absence of universally accepted testing methods and specialized equipment for assessing waste materials further complicates the accurate determination of strength parameters [33]. Given these challenges, site-specific investigations and the development of tailored testing protocols are essential for obtaining reliable geotechnical data on aged waste deposits. Understanding the evolving mechanical behavior of waste is crucial for ensuring safe and effective engineering solutions in landfill redevelopment projects.

The scientific literature presents different conclusions regarding the impact of time on the strength parameters of municipal waste. According to Fang [34], the shear strength of waste increases significantly over time, suggesting that long-term consolidation and material stabilization enhance its load-bearing capacity.

This perspective contrasts with the findings of Jessberger and Kockel [8], who argue that waste strength parameters tend to decrease over time due to ongoing decomposition and the breakdown of structural integrity. The normalized cone penetration test (CPT) soil behavior type (SBT) chart by Robertson [31] was used to distinguish municipal solid waste (MSW) of different degradation levels, as presented by Gomes [35]. Waste aged between 1.5 and 3.7 years was predominantly classified as sand mixtures (51%), sands (32%), and silt mixtures (10%), while the more degraded waste, aged 4.3 to 9.1 years, was categorized as sands (41%), sand mixtures (32%), and gravelly to dense sand (17%). According to both of Robertson’s charts [31,36], McKnight [32] found that less degraded MSW exhibited characteristics similar to sandy silt to sand, whereas more degraded waste behaved comparably to clayey soils. Given these conflicting viewpoints, computations of overall geotechnical safety of the landfill body and slope stability analyses on landfills must consider multiple scenarios to account for the potential variability in subsoil behavior.

The research presented in the paper attempts to estimate the mechanical parameters of aged and decomposed municipal waste based on static CPT performed on restored landfill sites. CPT investigations conducted at three closed municipal solid waste landfill sites were analyzed. During the investigations, cone resistance (q_c_) and sleeve friction (f_s_) were recorded. These values were used to interpret the geotechnical parameters of the investigated waste. The analyzed material was classified as having properties similar to unconsolidated or cohesive soils, depending on the results obtained from CPT (mainly based on the friction ratio R_f_). The aim was to confront the conclusions and the findings that the researchers of the published literature came up with. The proposed research is unique in its scale since the data come from reports conducted on three separate representative landfills, using the same equipment and the same crew who conducted the tests and interpreted the parameters themselves. Such an approach eliminates the possibility of receiving biased results due to using different investigation approaches, different rigs and instruments, and random work executors and labor.

## 2. Materials and Methods

### 2.1. Study Sites

The present study provides an overview of the scope and findings of a geotechnical investigation conducted on 3 major landfills located in central Poland, as presented in Figure 3.

All the landfills are located near the capital, where municipal solid waste has been landfilled since the 1960s. Two of them are considered the biggest MSW landfills in Poland. The authors of the paper have been involved in supervising all the landfill sites during their exploitation phase and reclamation process. Those sites served not only municipal purposes but were also considered as experimental sites for the investigation of highly complex physical, chemical, and biological mechanisms throughout the operational years. The site instigation conducted at those sites performed at different stages allowed for gathering representative in situ testing data that was included in the present study.

The first investigated site is the Radiowo landfill. The landfill, located near Warsaw, Poland, has been a key waste disposal site for the region since 1961. Initially receiving unsorted municipal waste, its use expanded due to capital urban growth. However, poor construction led to environmental challenges, including groundwater contamination, methane emissions, and stability issues, particularly landslides in the 1980s and 1990s. The landfill covers approximately 17.3 ha and reaches 60 m above ground level. The site is located on wetlands with a subsoil primarily composed of cohesive soils, interbedded with non-cohesive layers down to 10 m. The groundwater level is shallow (0–1.0 m), increasing the risk of leachate pollution. Since 1992, only non-composted waste from the Radiowo composting plant, including plastics, textiles, metals, and glass, has been disposed of. The organic matter content is below 4%, distinguishing older mixed municipal waste from the fresher, processed waste layers. In response to stability concerns, reclamation efforts from 1999 to 2017 included re-engineering slopes, reinforcing landfill structures, and installing drainage and degassing systems. A bentonite cut-off wall was constructed to limit groundwater contamination in 2000. Remedial works also introduced compost layers for vegetation development, aiming to mitigate erosion and enhance site stability. Public opposition has grown due to odor and environmental concerns, prompting stricter regulations. The landfill was closed in 2018, with ongoing discussions about repurposing the site for sustainable uses.

The second studied site is the Łubna landfill. The landfill is located 35 km south of Warsaw. It has been operating since 1978 until its closure in 2011. Covering an area of about 22 ha and reaching a height of nearly 60 m, it served as the primary disposal site for municipal waste from Warsaw, with peak loads of up to 2500 Mg per day in the late 1990s. Initially, waste disposal occurred in an unprepared wetland area, leading to severe environmental contamination. The subsoil of the landfill consists of non-cohesive sands and muds (2–15 m thick) overlying boulder and lacustrine clays. The groundwater table is shallow, at depths of 0.1 to 1.8 m. Before remediation efforts, the leachate from the unprotected landfill bottom infiltrated the first aquifer, affecting groundwater quality. In response, a cut-off wall (constructed in 1998), leachate drainage (1997–1998), and degassing systems were implemented. The bentonite vertical barrier 5.5–17 m deep effectively reduced leachate migration. The landfill is located within the Warsaw Plain, in a geomorphological zone characterized by a postglacial plateau and fluvial valleys. The area includes deposits from various glacial periods, with impermeable clay layers protecting deeper groundwater sources. However, shallow sand deposits allow infiltration of contaminants. Drainage ditches initially directed leachate toward local rivers (before establishing specific legal requirements in 1993), but later, the leachate was utilized in treatment facilities. The landfill site is surrounded by forests, grasslands, and agricultural fields, with former clay pits in the vicinity, some of which were illegally filled with waste. Continuous monitoring and maintenance of the leachate and gas extraction systems, as well as vegetation management, remain key aspects of post-closure environmental protection efforts until today.

The third studied site was the Zakroczym landfill. This is a non-hazardous and inert waste landfill, located in Zakroczym, Masovian Voivodeship, Poland, 35 km from the center of Warsaw. Covering an area of 6 hectares, the landfill was originally divided into three zones: the actively exploited southern section, the closed eastern section, and the reclaimed western section, which spans 1.5 ha. The restored western section was in operation from 1997 to 2011, during which 309 Mg of mixed municipal waste was disposed of. At the time of closure, the landfill crest reached an elevation of 116 m above sea level. Reclamation efforts began in 2012 and were completed by 2017, ultimately raising the landfill’s final elevation to 118 m above sea level. The restored section has a height of 8 m, with the total thickness of the waste layer reaching 12 m. The reclamation process involved a series of engineering and environmental remediation measures, including reinforcement of the landfill crown and slopes to ensure structural stability. The application of a leveling layer to create a uniform surface, installation of a 1 m thick clay capping layer, which serves as a protective barrier to minimize leachate infiltration, placement of a humus-based reclamation layer to support vegetation growth and enhance ecological restoration were implemented. Additionally, a biogas collection system was implemented as part of the landfill’s environmental management strategy. Several biogas extraction wells were installed, with three of them integrated into the landfill monitoring network to assess the composition and emissions of landfill gases regularly. This monitoring system is critical in ensuring compliance with environmental regulations and mitigating potential risks associated with landfill gas emissions.

### 2.2. Site Investigation Methods Used in Study for Determination of Mechanical Parameters of MSW

Due to the complex structure and diversity of processes influencing the formation of the waste profile within the study area, a series of tests was performed to determine the physical and mechanical parameters of waste starting from the top of the landfill body. Tests for evaluating the mechanical parameters of waste for strength analysis included morphological analysis, boreholes down to 5 m depth of the landfill, and CPT static tests with maximum depth down to a maximum of 20 m below the surface of the landfill crest. The schematic, graphical presentation of CPT results for waste along the landfill body is given in Figure 4, where a typical geological cross-section for the Łubna landfill is also presented.

The CPT involved pushing a cone penetrometer (van den Berg, Heerenveen, The Netherlands) with an integrated load cell into the ground at a controlled rate. The cone resistance (q_c_) in waste deposits is typically lower than in natural soils due to the high organic content and variable composition. Studies [8,37] have shown that q_c_ values in municipal solid waste (MSW) landfills range between 0.5 MPa and 5 MPa, depending on the degree of decomposition and compaction. The sleeve friction (f_s_) to cone resistance (q_c_) ratio (friction ratio, R_f_) helps distinguish organic-rich waste from inorganic components. The compressibility of waste is a key concern for landfill design. High organic matter content leads to significant long-term settlement due to biodegradation. Robertson and Cabal [36] suggested that empirical correlations developed for soil (e.g., q_t_-based correlations) require modification for waste due to its highly variable structure. Large settlements in MSW landfills have been linked to CPT parameters, and some models incorporate q_c_ values into settlement predictions.

Despite its advantages, CPT interpretation in waste faces a number of challenges. Waste deposits contain varying fractions of biodegradable and non-biodegradable materials, making direct interpretation difficult. Non-soil inclusions like plastic, wood, and metal can cause erratic q_c_ readings and equipment damage. Over time, the mechanical properties of waste change, requiring time-dependent interpretation models [38,39,40].

In the present case study, static CPT probing was performed using a Begemann mechanical tip (van den Berg, Heerenveen, The Netherlands) in accordance with the requirements of the EC7 standards [3]. The site investigation work performed at the landfill are documented in Figure 5. While pressing the probe tip, the values of cone pressing resistance (q_c_) and sleeve friction (f_s_) were determined. These values were used to identify geotechnical layers and determine the condition of waste behaving as non-cohesive soil (compaction index I_D_) and cohesive (liquidity index I_L_), as well as to determine the strength parameters of the waste (S_u_) for cohesive material and ϕ for non-cohesive materials occurring in the landfill body profile.

The geotechnical parameters were calculated using the empirical correlations as follows:

Liquidity index I_L_:I_L_ = A − 0.5 log (q_c_ − σ′_v0_)(1)
where q_c_ is the tip resistance, σ′_v0_ is the vertical effective geostatic stress, and A is the soil type-dependent factor (A = 0.32).

Relative density I_D_:I_D_ = 0.709 log q_c_ − 0.165(2)

Undrained shear strength for cohesive soil (S_u_):S_u_ = (q_c_ − σ_v0_)/*N_k_*(3)
where *N_k_* is the cone factor (*N_k_* = 20), and σ_v0_ is the total overburden pressure.

Internal friction angle φ [36]:φ = 0.125 I_D_ + B(4)
where B is the soil type-dependent factor (B = 28).

## 3. Results and Discussion

### 3.1. Mechanical Parameters of MSW Based on CPT Profiles

Figure 6, Figure 7 and Figure 8 show the results of the CPTs as a distribution of cone resistance q_c_, sleeve friction f_s_, and friction ratio R_f_ along the investigated depth for all three investigated landfills. Figure 9, Figure 10 and Figure 11 show the values of the compaction index I_D_ and the angle of internal friction ϕ, as well as the liquidity index I_L_ and shear strength S_u_, in the investigated profiles, calculated based on the testing results for all the analyzed landfills. The results of the cone penetration tests (CPTs) conducted on landfill sites provide valuable insights into the subsurface conditions, material composition, and mechanical behavior of the waste mass. The data collected from the tests highlight variations in soil resistance, stratification, and compaction across different sections of the landfill.

The CPT data for all analyzed landfills indicate significant heterogeneity in soil resistance values, reflecting the diverse nature of the landfill materials. Penetration resistance varied from low values in loosely compacted waste layers to high values in areas containing denser soil or construction debris. Sharp transitions in resistance profiles suggest the presence of distinct layers, likely resulting from differential compaction and deposition over time.

The results of geological boreholes and geotechnical tests allow us to conclude that the sublayers present complex geotechnical conditions, as they contain cohesive soil covering the landfill and municipal waste of various origins and in various conditions. In the top layer of the subsoil (capping of the landfill) down to 1.2–2.0 m, there are cohesive soils in the form of clayey sands and sandy clays, sandy silt, and clays. Below this layer, a transition to municipal waste of varying compositions, densities, and decomposition stages is observed, which significantly affects the geomechanical behavior of the subsoil.

As shown in the results, these deposits are mostly medium compacted, layered by stiff material, and thus present relatively good geotechnical conditions. Beneath the capping layer, the landfill comprises well-compacted, mostly mineralized waste material. The composition of this waste varies depending on location, a typical observation for landfills of this type. The analysis of the CPT results, particularly the friction ratio R_f_, indicates that these waste exhibit characteristics of both non-cohesive soils (R_f_ ≤ 2.2) and cohesive soils (R_f_ > 2.2). Waste classified as cohesive material is predominantly in a stiff state (I_L_ = 0.05–0.15), with some areas exhibiting a medium stiff (I_L_ < 0) or plastic consistency (CPT-3; I_L_ = 0.25–0.35). Non-cohesive waste materials display moderate compaction (I_D_ = 0.3–0.6) and well-compacted conditions (I_D_ = 0.7–0.8) in specific locations (CPT-4, CPT-5 for Zakroczym landfill), though occasionally they appear in a loose state (I_D_ < 0.3).

Moisture levels across the test locations influenced the recorded resistance values, with higher moisture content correlating to lower penetration resistance. This indicates a potential for higher compressibility in saturated regions, which could lead to differential settlement in the landfill over time. The presence of organic waste further exacerbates compressibility due to ongoing decomposition, leading to gradual subsidence and void formation within the landfill mass. Additionally, localized variations in moisture levels suggest differential drainage conditions, which could impact leachate migration and the overall hydrodynamic balance of the landfill.

Several test locations exhibited abrupt reductions in resistance, suggesting the presence of voids or decomposed organic material. These anomalies can compromise the structural integrity of the landfill and pose challenges for future land use planning. The values of cone and sleeve friction obtained from CPTs for the analyzed municipal waste fall within the range of 1–8 MPa. Zones of sharp increases in resistance indicate the presence of very high-stiffness materials, which, in extreme cases, may limit the investigation depth. The presence of gas pockets, likely due to anaerobic degradation, also raises stability concerns and requires consideration in landfill gas management strategies. The variations observed in penetration resistance and soil layering directly affect landfill stability. Areas with lower resistance values may require reinforcement or controlled settlement monitoring to prevent structural failures.

Additionally, detecting heterogeneous material composition highlights the importance of ongoing waste compaction efforts to enhance landfill performance. Static geotechnical test results revealed significant variations in strength parameters due to the differing mechanical behaviors of cohesive and non-cohesive waste. The values of shear strength in undrained conditions from CPTs ranged from 50 to 350 kPa, while the internal friction angle varied between 28° and 39°. The full interpretation of the results is given in Table 1. All the observations confirm the differences in the waste degradation level and different moisture content, as well as the compaction index of the analyzed waste layers of the landfill body.

### 3.2. Classification of Waste Materials Based on Cone Resistance and Friction Ratio Results

Analyzing the distribution of the values presented in the nomograms in Figure 12 and Figure 13 suggests that municipal waste behaves similarly to soil, covering a range from clay to sand, and often exceeding waste classification ranges in the existing literature.

When analyzing the previous testing profiles, cohesive waste layers extending to a depth of 5 m exhibit a hard-plastic state (I_L_ = 0.06) with a shear strength of S_u_ = 190 kPa. Below this depth, these materials transition to a very low plasticity state (I_L_ = −0.02) with a shear strength of S_u_ = 260 kPa. Notably, from a depth of 5 m, an increase in the shear strength of these cohesive waste is observed. In contrast, non-cohesive waste materials exhibit the opposite trend. Down to a depth of 3.5 m, these materials are in a medium compacted state (I_D_ = 0.60), but below this depth, they transition to a loose to medium compacted state (I_D_ = 0.33). The average internal friction angle in the upper 3.5 m is φ = 36°, decreasing to φ = 32° at greater depths. A clear reduction in the internal friction angle beyond 3.5 m is observed, which has implications for stability assessments and foundation design for planned infrastructure projects. The accurate interpretation of measured parameters, such as cone resistance (q_c_), sleeve friction (f_s_), and friction ratio (R_f_), at various depths enables the use of existing nomograms.

These nomograms are aimed at classifying landfill materials and providing empirical estimates of their strength parameters. The data collected from all the landfills were gathered and presented in the form of nomograms, as presented in Figure 12 and Figure 13. To further assess landfill conditions, supplementary investigations such as seismic testing, borehole sampling, and long-term settlement monitoring should be considered.

These complementary methods can refine our understanding of landfills’ geotechnical characteristics and inform the development of sustainable management strategies. Additionally, incorporating numerical modeling of landfill settlement and stability can aid in predicting future behavior and optimizing design solutions. Regular gas and leachate monitoring should also be performed to mitigate potential environmental hazards. For the purposes of foundation design for photovoltaic panels, geotechnical layers were divided into exploration depth zones, each characterized by unified geotechnical parameters. The ground conditions of the analyzed profiles—relative density I_D_ for non-cohesive soils and plasticity index I_L_ for cohesive soils—along with grain size distributions, were adopted as criteria for distinguishing geotechnical layers.

### 3.3. Discussion on the Interpretation of the Obtained Research Results

To facilitate geotechnical interpretation, waste materials were classified into cohesive or non-cohesive groups based on the friction R_f_. However, it is important to note that geotechnical nomograms commonly used for soil classification may not provide precise interpretations for waste materials due to the lack of established guidelines in the literature. Based on the analyses, the authors confirm the McKnight [32] observation that less degraded MSW (located closer to the surface) exhibits characteristics similar to sandy silt to sand, whereas more degraded (at lower depths of landfills) waste behaved comparably to clayey soils.

However, when looking closely at the results, the authors state that there is a significant lack of very fine non-cohesive soil type waste fractions observed in all analyzed landfills. There is a clear gap marked in red in Figure 12 and Figure 13, which proves the lack of certain types of deposits that could be clearly classified as fine silty sands or fine silts. The “red gap” on the nomograms could be associated with a specific type of waste that was deposited in the past, the level of degradation, or different mineralization processes influencing the aging, decomposition, and eventually, the structure and shear strength.

For the investigated waste of friction ratio (R_f_) from the range 1.5–1.9 (“red gap”), cohesive fraction of waste was not observed at all, in contrast to McKnight’s observation. This specific range of R_f_ needs to by further studied, paying particular attention to the depth, year of deposition, and the level of degradation and type of waste falling in the specified range of friction ratio. Further analyses confirm the theory of Dixon and Jones [6] that the internal friction angle (ϕ) tends to decrease as waste decomposition progresses, specifically due to biodegradation over extended depths (longer mineralization time, significant aging effect). Whereas Feng’s [18] observation was confirmed in the present study, meaning that the shear strength of waste increases significantly over time, suggesting that long-term consolidation and material stabilization enhance its load-bearing capacity, the authors claim that such behavior only applies to cohesive soil alike waste in all three landfills at certain depths. Moreover, after detailed analyses of the results, the authors claim that, in contrast to Jessberger and Kockel [8], the observation that the shear strength parameters drop with the depth applies only for two out of three landfills and the behavior differs for cohesive and non-cohesive waste material.

In summary, the cone penetration tests have provided a detailed understanding of the landfill’s subsurface conditions, identifying key factors influencing stability and long-term performance. These findings can be used to guide future engineering decisions, and could help in establishing more precise and reliable constitutive models and powerful equations for determining mechanical parameters for such complex materials as municipal solid waste [42,43]. It will undoubtedly enhance landfill management practices and environmental monitoring efforts, ensuring the site remains structurally safe and suitable for planned developments.

## 4. Conclusions

Cone penetration tests (CPTs) conducted at three landfill sites reveal significant heterogeneity of waste materials, characterized by variations in cone and sleeve resistance and compaction. The upper 1.2–2.0 m layer generally consists of cohesive soils, underlain by municipal solid waste (MSW) of variable composition, density, and decomposition. This layered waste includes both cohesive and non-cohesive materials, each with distinct mechanical behaviors: cohesive waste exhibits lower undrained shear strength near the surface that increases with depth, while non-cohesive waste shows higher internal friction angles in upper layers that decrease deeper down the investigation profile. CPT data underscore the importance of waste compaction in maintaining landfill integrity and enabling reliable site reclamation work. The findings confirm that MSW can exhibit soil-like mechanical behavior, ranging from clay-like to sandy properties. Shear strength, internal friction angle (φ), and compaction indices are influenced primarily by moisture and degradation state. The results support the widely accepted trend of decreasing φ with progressive decomposition. However, the behavior is not uniform. Only two of the three landfills showed consistent shear strength reductions with depth, making the material variability and the influence of site-specific conditions even more complex to interpret. An absence of fine non-cohesive fractions (e.g., silty sands) was observed in all landfills, shown by a distinct “red gap” in the nomograms at friction ratio (R_f_) values between 1.5 and 1.9. This absence may reflect historical waste disposal practices or material degradation and requires further study. Interestingly, the CPT results across all sites lacked data points in this transitional R_f_ range. It is an unexpected outcome given the material diversity, which could have implications for future classification methods. The study proposes practical thresholds for classifying MSW based on R_f_ values: waste with R_f_ < 1.5 behaves as non-cohesive, while R_f_ > 1.9 suggests cohesive behavior. Unlike mineral soils, this boundary is not obvious, forming a transition zone. This range, absent from measured data in the study, suggests a diagnostic gap that future research could help define. Recognizing this boundary is crucial for applying correct N_kt_ values when interpreting CPT data and estimating shear strength. One of the strengths of the present research is its methodological consistency. All data were collected using the same CPT equipment, operated by the same trained crew, and analyzed through a uniform interpretative approach across three separate representative landfill sites. This uniformity enhances both the reliability and the internal validity of the results, reducing methodological variability that often hinders comparability in similar studies. The authors first identified a research gap, which is the absence of a standardized testing method for in situ waste characterization, and addressed it by demonstrating not only practical feasibility but also reproducibility under controlled conditions of CPT. While cone penetration testing proves to be a valuable tool for evaluating landfill stability based on mechanical parameters of waste, the authors acknowledge that CPT-derived data must be interpreted with caution when approaching reclamation work. To further validate the geotechnical parameters obtained, and to improve external validity, CPT should be complemented with supporting methods such as seismic testing, borehole sampling, and long-term settlement monitoring. The integration of site-specific data with numerical modeling will be critical in extending the applicability of these findings to diverse landfill contexts, ultimately contributing to the establishment of a more standardized and robust methodology for future research in this field.

## Figures and Tables

**Figure 1 materials-18-02130-f001:**
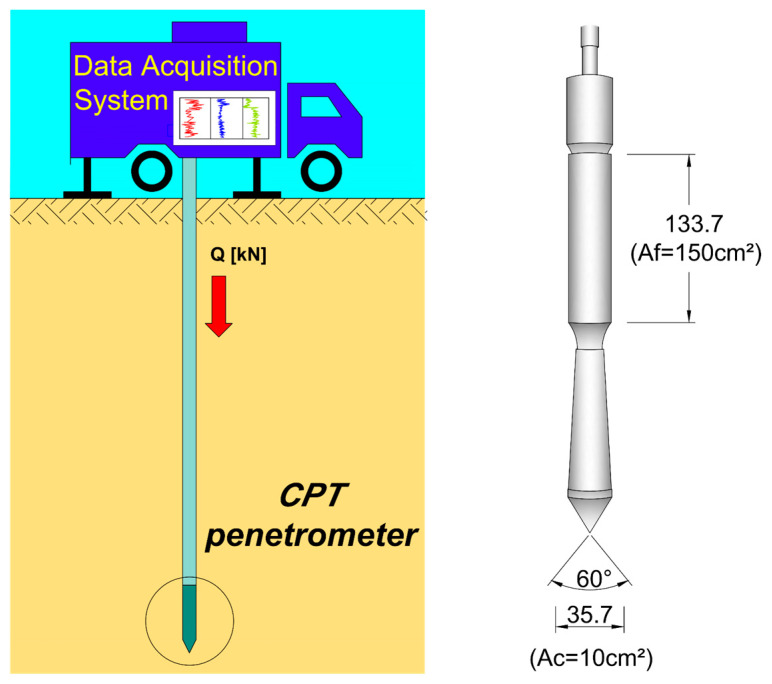
Cone penetration test procedure with cone detailed features.

**Figure 2 materials-18-02130-f002:**
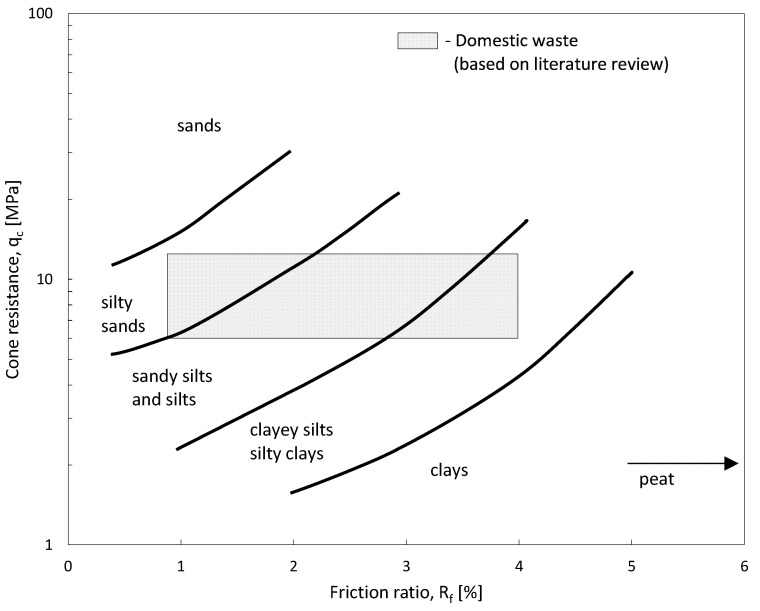
Nomogram for landfill waste parameters classification (modified after [11]).

**Figure 3 materials-18-02130-f003:**
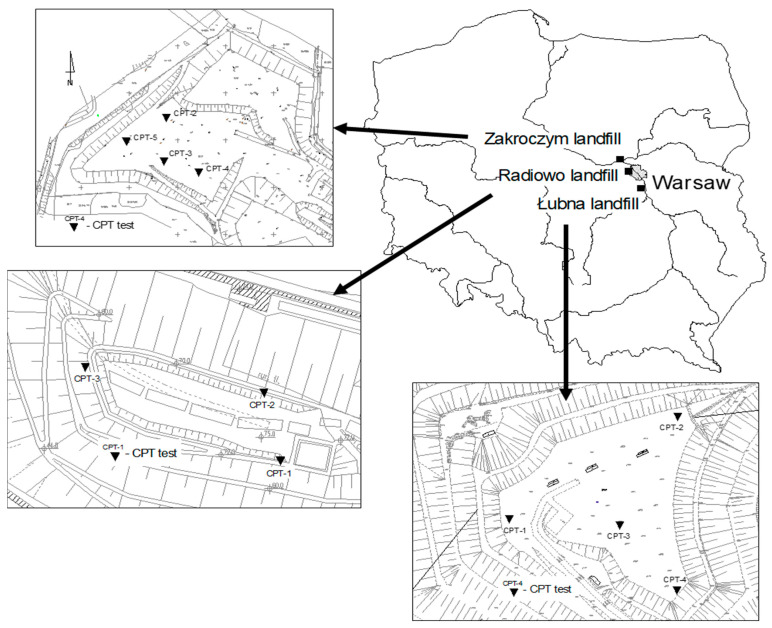
Landfills’ locations.

**Figure 4 materials-18-02130-f004:**
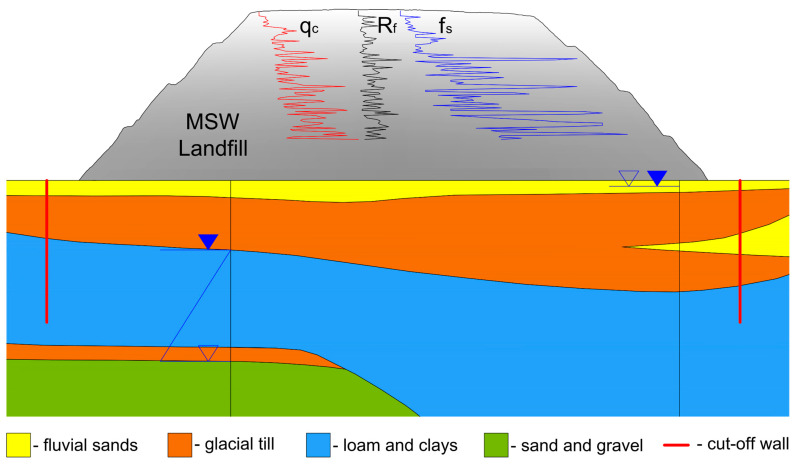
The schematic representation of the CPT results of waste materials building the landfill body on a typical geological cross-section.

**Figure 5 materials-18-02130-f005:**
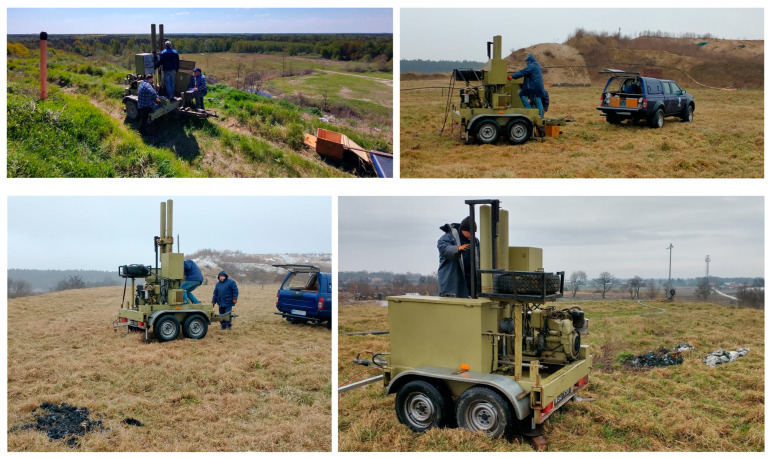
Site investigation work on Zakroczym and Łubna landfills’ crest using CPT platform.

**Figure 6 materials-18-02130-f006:**
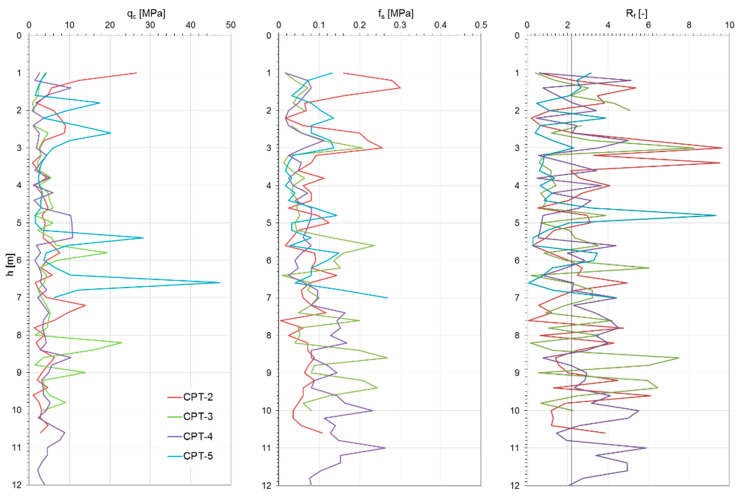
Zakroczym landfill CPT results.

**Figure 7 materials-18-02130-f007:**
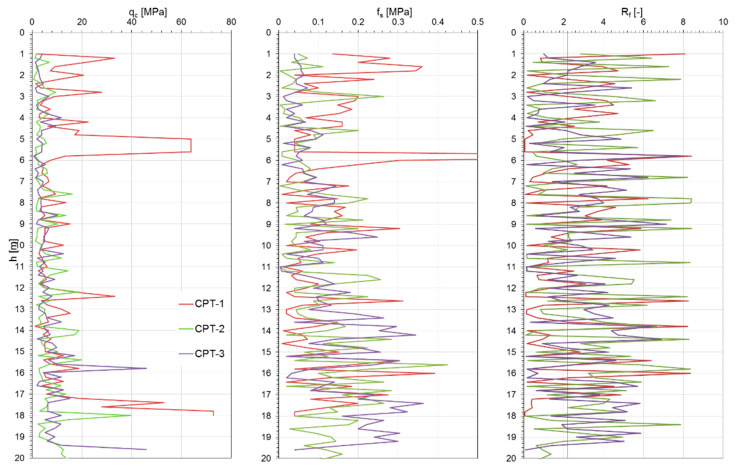
Radiowo landfill CPT results.

**Figure 8 materials-18-02130-f008:**
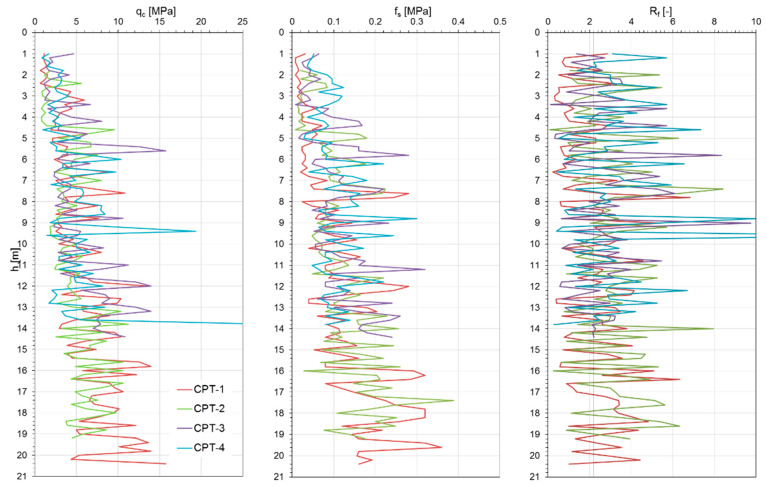
Łubna landfill CPT results.

**Figure 9 materials-18-02130-f009:**
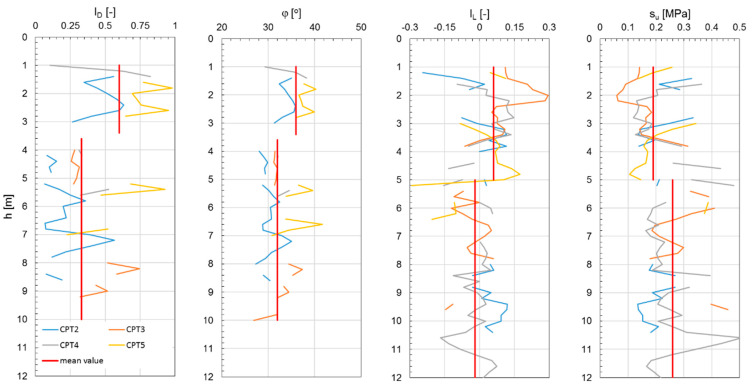
Investigated geotechnical parameters from CPTs performed on Zakroczym landfill.

**Figure 10 materials-18-02130-f010:**
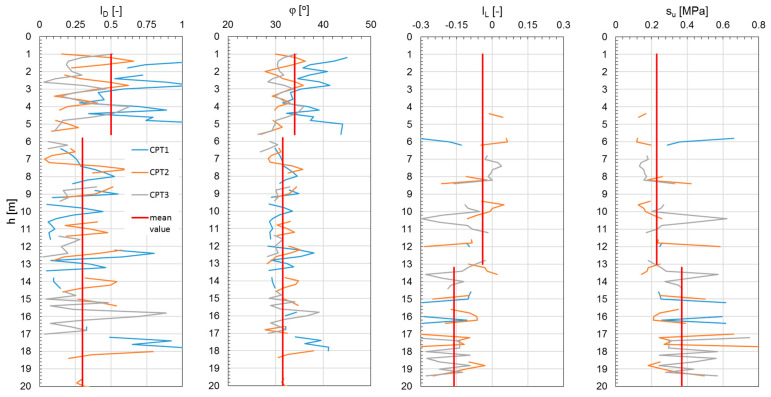
Investigated geotechnical parameters from CPTs performed on Radiowo landfill.

**Figure 11 materials-18-02130-f011:**
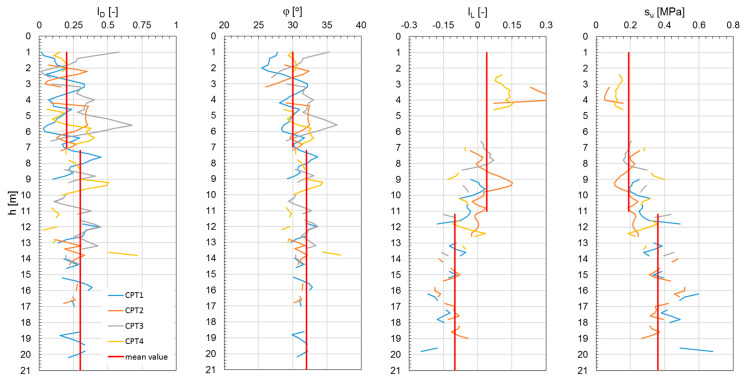
Investigated geotechnical parameters from CPTs performed on Łubna landfill.

**Figure 12 materials-18-02130-f012:**
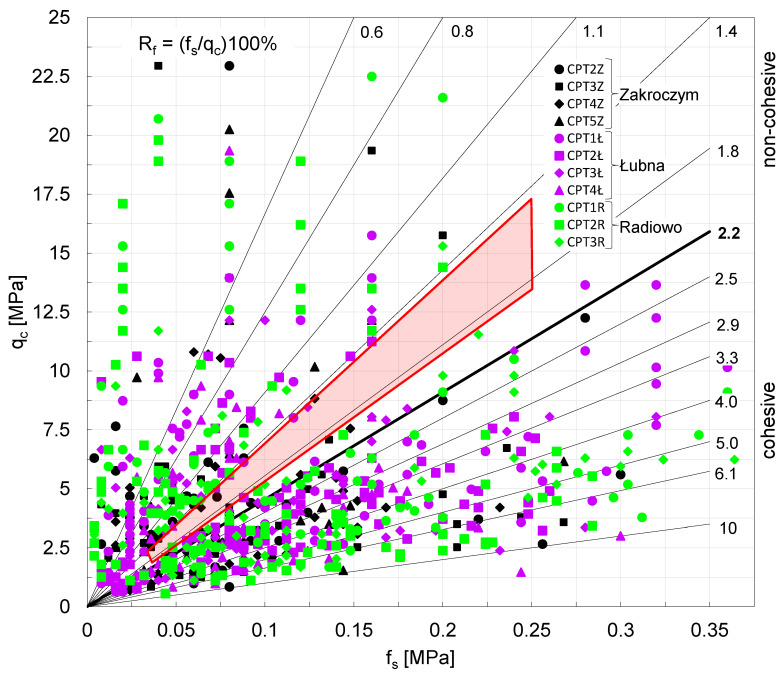
Results of CPT-based parameters for investigated waste plotted on nomograms for landfill waste parameters classification (“red gap” marks absence of certain type of fraction).

**Figure 13 materials-18-02130-f013:**
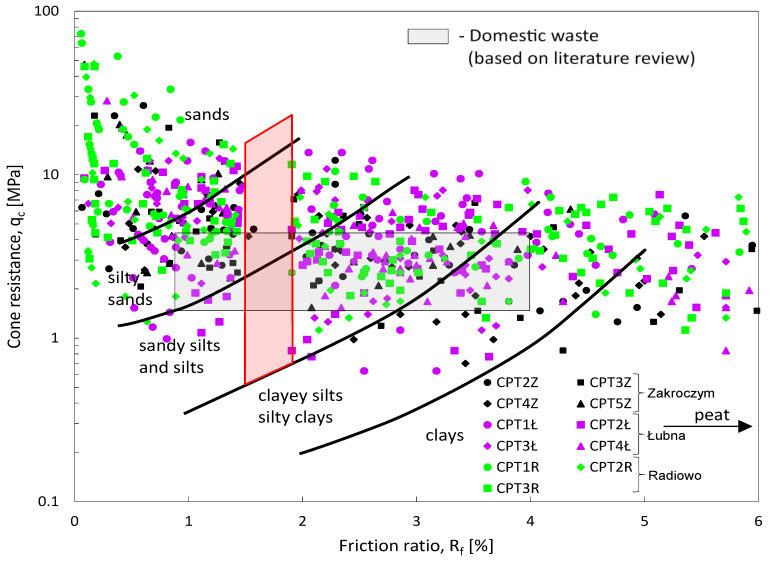
Results of CPT-based parameters for friction ratio and cone resistance of investigated waste referring to soil type classification acc. [41] (“red gap” marks absence of certain type of fraction).

**Table 1 materials-18-02130-t001:** Waste geotechnical parameters proposed based on CPTs for analyzed landfills.

Landfill	Description	Density Index I_D_ [-]	Liquidity Index I_L_ [-]	Angle of Friction φ [°]	Shear Strength Su [kPa]
**Zakroczym**	“cohesive” waste	-	<0	-	250–350
		-	0.05–0.15	-	100–200
		-	0.25–0.35	-	50–80
	“non-cohesive” waste	0.70–0.80	-	37–39	-
		0.40–0.60	-	34–36	-
		0.10–0.30	-	28–31	-
**Łubna**	“cohesive” waste	-	<0	-	250–350
		-	0.05–0.15	-	100–200
		-	0.25–0.35	-	50–80
	“non-cohesive” waste	0.70–0.90	-	37–40	-
		0.40–0.60	-	34–36	-
		0.10–0.30	-	28–31	-
**Radiowo**	“cohesive” waste	-	<0	-	200–400
		-	0.00–0.14	-	100–150
	“non-cohesive” waste	0.89–1.00	-	40–45	-
		0.69–0.80	-	37–39	-
		0.35–0.65	-	33–36	-
		0.10–0.30	-	28–32	-

## Data Availability

The raw data supporting the conclusions of this article will be made available by the authors on request.

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
