# Peer review of "Mechanical Strength of Waste Materials: A Cone Penetration Testing-Based Geotechnical Assessment for the Reclamation of Landfills"

_materials, 2025, doi:10.3390/ma18092130_

Round 1

Reviewer 1 Report

Comments and Suggestions for Authors

The manuscript presents CPT results conducted on three reclaimed landfill sites. CPT has been conducted as part of a geotechnical investigation to assess the ground condition. Authors have presented the raw CPT data and interpreted them to obtain several geotechnical parameters. Data presented is useful for designers and future researchers.

I have some major concerns about this manuscript that should be addressed before it can be recommended for publication.

  1. Titel: Why does it contain the word ‘sustainable’? You have used the CPT in the standard way. What is more sustainable about it? I suggest removing it. Also, it should be “Mechanical strength of waste materials.”
  2. Introduction is too long especially the paragraph from L80 to L135. Shorten and break them into small paragraphs. Use subheadings where appropriate to better organise ideas.
  3. Generally, paper contains long paragraphs and repetitive ideas which makes it cumbersome to read. For example, the ideas in L311 to L314 were discussed earlier as well. You can substantially shorten the manuscript by avoiding repetition.
  4. Revise the sentences in L304 to L307. The idea is not clear. Figure 4 only shows a typical cross section of a landfill, not the approach. Not even the heights of landfill, depth of CPT are marked on it. Colours used for fluvial sand and sand and gravel are very similar and hard to differentiate. At the moment, this figure is not very useful.
  5. Add subheadings to the discussion to organise ideas and to enhance readability.
  6. Conclusion is too long. This section should be a summary of key findings of the study.

Author Response

The authors would like to express their appreciation for such detailed reviews and valuable comments, which undoubtedly improved the scientific quality of the corrected manuscript.

The authors would like to confirm that the manuscript was corrected according to all suggestions from the Reviewer. The entire text was carefully reviewed and corrected where required.

The answers to the Reviewer comments were submitted via the susy.mdpi website and are also attached to the present Cover Letter. The authors would like to assure you that we did our best to address all the comments and include the answers in the main text where applicable. The manuscript was proofread, carefully reviewed, double-checked and corrected. For all specific answers to the given comments and remarks, please refer to ANSWER TO COMMENTS .pdf file or see below:

Reviewer 1

Specific comment 1:

Titel: Why does it contain the word ‘sustainable’? You have used the CPT in the standard way. What is more sustainable about it? I suggest removing it. Also, it should be “Mechanical strength of waste materials.”

Answer:

The authors used the term “sustainable” to reflect the reclamation approach adopted for the landfills, where green energy facilities were proposed. However, since the paper focuses on investigating the waste material properties more than on the reclamation works, the authors agree that the title should be modified. As suggested by Reviewer 1, the title is now Mechanical strength of waste materials: A CPT-Based Geotechnical Assessment for Landfills’ Reclamation

Specific comment 2:

Introduction is too long especially the paragraph from L80 to L135. Shorten and break them into small paragraphs. Use subheadings where appropriate to better organise ideas.

Answer:

The authors agree that some of the information could be presented in a more concise manner. The authors revised and rewrote the text, and also, as suggested by the Reviewer, there are now 3 new paragraph headings added to make the manuscript's structure more organised and scientifically sound. The repetitions were eliminated, and the introduction was shortened by 367 words. Please refer to the corrected copy to see the changes that could be followed in tracking mode.

Specific comment 3:

Generally, paper contains long paragraphs and repetitive ideas which makes it cumbersome to read. For example, the ideas in L311 to L314 were discussed earlier as well. You can substantially shorten the manuscript by avoiding repetition.

Answer:

The manuscript was restructured, and major repetitions were eliminated. All the sections were revised, including methodology, discussion and conclusion. The authors hope the present form will make the manuscript easier to follow and reader-friendly. For changes, please see the revised copy. The present actual content was now shortened by approximately 700 words, trying to maintain still the idea of the research approach and its outcomes. The repetitions removed are: l. 198-203; 240-244; 253-254; 357-363; 364-372 (track changes copy in .docx file). The authors would like to express their appreciation for pointing that out. We hope that the present form of the manuscript will make the reading flawless.

Specific comment 4:

Revise the sentences in L304 to L307. The idea is not clear. Figure 4 only shows a typical cross section of a landfill, not the approach. Not even the heights of landfill, depth of CPT are marked on it. Colours used for fluvial sand and sand and gravel are very similar and hard to differentiate. At the moment, this figure is not very useful.

Answer:

The sentences were revised and rewritten, and is now:

“Tests for evaluating the mechanical parameters of waste for strength analysis included morphological analysis, boreholes down to 5 m depth of the landfill, and CPT static tests with maximum depth down to a maximum of 20 m below the surface of the landfill crest. The schematic, graphical presentation of CPT results for waste along the landfill body is given in Figure 4, where a typical geological cross-section for Łubna landfill is also presented.”  l. 346-351 in marked copy of the manuscript (.docx file with tracking changes).

Figure 4 was modified and replaced. The colours representing different soil layers were distinguished in contrasting shades. The caption was also rewritten to reflect the figure's content more precisely. Using scale and axis labels for dimensions would make the figure more challenging to read and understand. For that reason, the caption of Fig.4 was changed to “The schematic presentation of CPT results of waste materials building the landfill body on a typical geological cross-section.” All the changes can be tracked in the corrected copy of the manuscript.

Specific comment 5:

Add subheadings to the discussion to organise ideas and to enhance readability.

Answer:

As suggested, the 3 additional subheadings were added to the Results and Discussion section to provide a better understanding and make the text easier to follow. These are:

3.1 Mechanical parameters of MSW based on CPT profiles

3.2 Classification of waste materials based on cone resistance and friction ratio results

3.3 Discussion on the interpretation of the obtained research results

The authors agree that such a form will make the content more reader-friendly. For details, please refer to a corrected copy of the manuscript.

Specific comment 5:

Conclusion is too long. This section should be a summary of key findings of the study.

Answer:

The section was significantly reduced (by 140 words). The conclusions are now focused only on crucial findings and outcomes of the research. Please see l. 615-658 (marked copy with track changes in .docx file)

The authors would like to express their great appreciation for providing such detailed reviews and for all the Reviewer efforts put in improving the manuscript for better scientific sound and presentation. The authors would like to ensure that all the corrections were implemented and all general and specific comments were addressed in the revised copy of the manuscript.

With best regards,

Authors of the manuscript

Reviewer 2 Report

Comments and Suggestions for Authors

This manuscript employs the Cone Penetration Testing (CPT) method to investigate the mechanical properties at three different municipal solid waste landfill sites in Poland. It holds a certain guiding significance from an engineering practice perspective and presents well-executed experimental work. The findings are expected to be beneficial for future studies on landfill stability. Some recommendations are listed below:

  1. Please address a few minor issues:
    • Line 325: The “c” in “qc” should be in subscript.
    • Line 457: Please check the width of the table border.
  2. In this manuscript, the authors identify two research gaps. The first gap is the lack of a standardized testing method for determining geotechnical parameters of in-situ waste fill. In addressing this, the authors applied the same CPT equipment and methodology across all three landfill sites, which supports comparability of the results. It is recommended that the authors further discuss the reliability and validity of the CPT-derived data, particularly regarding its applicability in future research.

Author Response

The authors would like to express their appreciation for such detailed reviews and valuable comments, which undoubtedly improved the scientific quality of the corrected manuscript.

The authors would like to confirm that the manuscript was corrected according to all suggestions from the Reviewer. The entire text was carefully reviewed and corrected where required.

The answers to the Reviewers’ comments were submitted via the susy.mdpi website and are also attached to the present Cover Letter. The authors would like to assure you that we did our best to address all the comments and include the answers in the main text where applicable. The manuscript was proofread, carefully reviewed, double-checked and corrected. For all specific answers to the given comments and remarks, please refer to ANSWER TO COMMENTS in .pdf file attached or see below:

Reviewer 2

Specific comment:

Please address a few minor issues:

Line 325: The “c” in “qc” should be in subscript.

Line 457: Please check the width of the table border.

 Answer:

The index was corrected, and the table format now meets the journal's requirements. Thank you for pointing that out.

Specific comment:

In this manuscript, the authors identify two research gaps. The first gap is the lack of a standardized testing method for determining geotechnical parameters of in-situ waste fill. In addressing this, the authors applied the same CPT equipment and methodology across all three landfill sites, which supports comparability of the results. It is recommended that the authors further discuss the reliability and validity of the CPT-derived data, particularly regarding its applicability in future research.

Answer:

The authors would like to express their appreciation for such positive feedback. The authors believe that indeed a key strength of this research lies in its methodological consistency. The uniformity enhances both the reliability and the internal validity of the results, reducing methodological variability that often disrupts comparability in similar studies. To enhance and improve the scientific value of the manuscript the Authors addressed the comment in Conclusion section. Please see l. 642-658 (.docx file with track changes copy):

One of the strengths of the present research is its methodological consistency. All data were collected using the same CPT equipment, operated by the same trained crew, and analyzed through a uniform interpretative approach across three separate representative landfill sites. This uniformity enhances both the reliability and the internal validity of the results, reducing methodological variability that often hinders comparability in similar studies. The authors first identified a research gap, which is the absence of a standardized testing method for in-situ waste characterization, and addressed it by demonstrating not only practical feasibility but also reproducibility under controlled conditions of CPT. While the cone penetration test proves to be a valuable tool for evaluating landfill stability based on mechanical parameters of waste, the authors acknowledge that CPT-derived data must be interpreted with caution when approaching reclamation works. To further validate the geotechnical parameters obtained, and to improve external validity, CPT should be complemented with supporting methods such as seismic testing, borehole sampling, and long-term settlement monitoring. The integration of site-specific data with numerical modeling will be critical in extending the applicability of these findings to diverse landfill contexts, ultimately contributing to the establishment of a more standardized and robust methodology for future research in this field.”

The authors would like to express their great appreciation for providing such detailed reviews and for all the Reviewers’ and the Editor's efforts put in improving the manuscript for better scientific sound and presentation. The authors would like to ensure that all the corrections were implemented and all general and specific comments were addressed in the revised copy of the manuscript.

With best regards,

Authors of the manuscript
